# Gold Nanoparticle-Decorated Bi₂S₃ Nanorods and Nanoflowers for Photocatalytic Wastewater Treatment

**Njemuwa Nwaji [1,\*], Eser Metin Akinoglu [1,2] and Michael Giersig [1,3]**

1   International Academy of Optoelectronics at Zhaoqing, South China Normal University, Liyuan Street, Zhaoqine 526238, China; e.a@fu-berlin.de or eser.akinoglu@unimelb.edu.au (E.M.A.); giersig@zedat.fu-berlin.de or mgiersig@ippt.pan.pl (M.G.)
2   ARC Centre of Excellence in Exciton Science, School of Chemistry, University of Melbourne, Parkville, VIC 3010, Australia
3   Institute of Fundamental Technological Research, Polish Academy of Sciences, 02-106 Warsaw, Poland
\*   Correspondence: njemuwa.nwaji@zq-scnu.org; Tel.: +86-18024972217

**Abstract:** Colloidal synthesis of photocatalysts with potential to overcome the drawback of low photocatalytic efficiency brought by charge recombination and narrow photo-response has been a challenge. Herein, a general and facile colloidal approach to synthesize orthorhombic phase Bi₂S₃ particles with rod and flower-like morphology is reported. We elucidate the formation and growth process mechanisms of these synthesized nanocrystals in detail and cooperate these Bi₂S₃ particles with metallic gold nanoparticles (AuNPs) to construct heterostructured photocatalysts. The unique properties of AuNPs featuring tunable surface plasmon resonance and large field enhancement are used to sensitize the photocatalytic activity of the Bi₂S₃ semiconductor particles. The morphology, structure, elemental composition, and light absorption ability of the prepared catalysts are characterized by (high-resolution) transmission electron microscopy, scanning electron microscopy, X-ray diffraction spectroscopy, X-ray photoelectron spectroscopy, and UV–vis absorption spectroscopy. The catalysts exhibit high and stable photocatalytic activity for the degradation of organic pollutants demonstrated using rhodamine B and methyl orange dyes under solar light irradiation. We show that the incorporation of the AuNPs with the Bi₂S₃ particles increases the photocatalytic activity 1.2 to 3-fold. Radical trapping analysis indicates that the production of hydroxyl and superoxide radicals are the dominant active species responsible for the photodegradation activity. The photocatalysts exhibit good stability and recyclability.

**Keywords:** Bi₂S₃; nanoflower; nanorod; photocatalysis; heterostructures; AuNPs

## 1. Introduction

There has been a rising research interest aimed at solving the current energy and environmental challenges. The environmental pollution arising from pervasive commercial dyes has posed a great environmental problem [1,2]. The dye effluents from textile mills have been the largest discharge of dyes into the environment and confers an acute problem for municipal waste treatment facilities. A significant degradation of organic dye pollutant has not been achieved using technologies involving physical or biological treatments [3,4]. Alternatively, solar energy, which is the most promising clean and renewable energy source, is considered to power these processes. The radiative energy of the sun can be harvested through light absorption in semiconductors. Such semiconductors in particle form are good photocatalysts and present an alternative treatment modality to successfully oxidize notorious organic dye pollutants existing in wastewater.

The development of cost-effective, high-performance, and earth abundant photocatalysts is an effective means to address the issue of solar energy conversion to chemical energy. Despite these advantages of photocatalytic energy conversions using diverse chalcogenides semiconductors, the efficiency is limited mainly due to fast charge recombination and the

requirement for semiconductors with wide band gap to deliver the required potential differences for relevant redox reactions. On the other hand, narrow band gap semiconductors absorb a larger spectrum of light. An ideal semiconductor photocatalyst for water should have a band gap of at least 1.23 eV, and its conduction band and valence band positions should straddle the redox levels for the water oxidation and reduction reactions, respectively [5–7]. Many and varied metal chalcogenides have been studied for the hydrogen evolution reaction and the degradation of organic pollutants due to their large absorption coefficient [8–10]. Among diverse metal sulfides, $Bi_2S_3$ has been widely studied as an efficient absorber of solar light due to its suitable narrow band gap (1.3–1.7 eV), suitable band edge potentials, non-toxic nature, good incident to photon conversion efficiency, as well as high absorption coefficient ($10^4$–$10^5$ cm$^{-1}$), which is also favorable for applications in optoelectronic devices, thermoelectric devices, nonlinear optical devices, optical modulators, and sensors [11–16]. However, the photocatalytic efficiency of $Bi_2S_3$ is poor due to rapid electron–hole recombination, photo-corrosion, and slow hole transfer kinetics at the $Bi_2S_3$/electrolyte interface [17,18]. Many and diverse heterostructured nanomaterials with different morphology such as rods [19,20], flowers [21], wires [22] and cubes [23] have been reported in the field of photocatalysis. The loading of noble metals such as Au, Ag, and Pt as a catalyst and the construction of semiconductor heterostructures from more than one semiconductor for charge separation at their heterojunction are strategies to improve the harvesting of photoexcited charge carriers [24,25]. Although Au-$Bi_2S_3$ heteronanostructures were reported for diverse applications [26–28], the synthetic procedures either pose more difficulty or the prepared materials showed low photocatalytic activity. For example, a seed-mediated approach was employed by Ma et al. [21] to prepare Au core and $Bi_2S_3$ shell heterostructures. Apart from the multi-stage synthesis as well as the use of toxic reagent like sodium borohydride, hexadecyltrimethylammonium bromide, the formed core-shell Au@$Bi_2S_3$ showed low photocatalytic activity of 62% toward the degradation of RhB. Mana et al. [22] and Li et al. [23] prepared $BiS_3$ nanorods followed by forming Au-$Bi_2S_3$ heterostructures. Their synthetic approach used both solvents and a sulfur source with different reagents such as Oleyamine, oleic acid octadecene, and triphosphine oxide, which poses difficulty in purification.

In this work, we report a facile one-pot method to synthesize rod and flower-like shaped $Bi_2S_3$ nanocrystals, which we abbreviate as BNR and BNF, respectively, utilizing 1-docanethiol (DDT) both as solvent and sulfur source (Figure 1), making our synthetic approach green with low cost. A strategic rational approach was employed to decorate the surface of these nanocrystals with metallic gold nanoparticles to form heterostructured nanocomposites abbreviated as BNR-Au and BNF-Au, respectively. In these systems, enhancement in photocatalytic activity arising from the large local field and Mie scattering induced absorption for semiconductor $Bi_2S_3$ is expected. The plasmonic Au cocatalyst nanoparticles act as photosensitizers due to their unique localized surface plasmon resonance (LSPR) effect. Furthermore, plasmon-induced hot electrons that are generated from the non-radiative decay of plasmons inside the AuNPs could be the source for additional photogenerated charge carriers available to do photochemistry. Additionally, the strong plasmon-induced electric field has the potential to increase the absorption cross-section of the semiconductor, which may also affect the exciton dynamics [26].

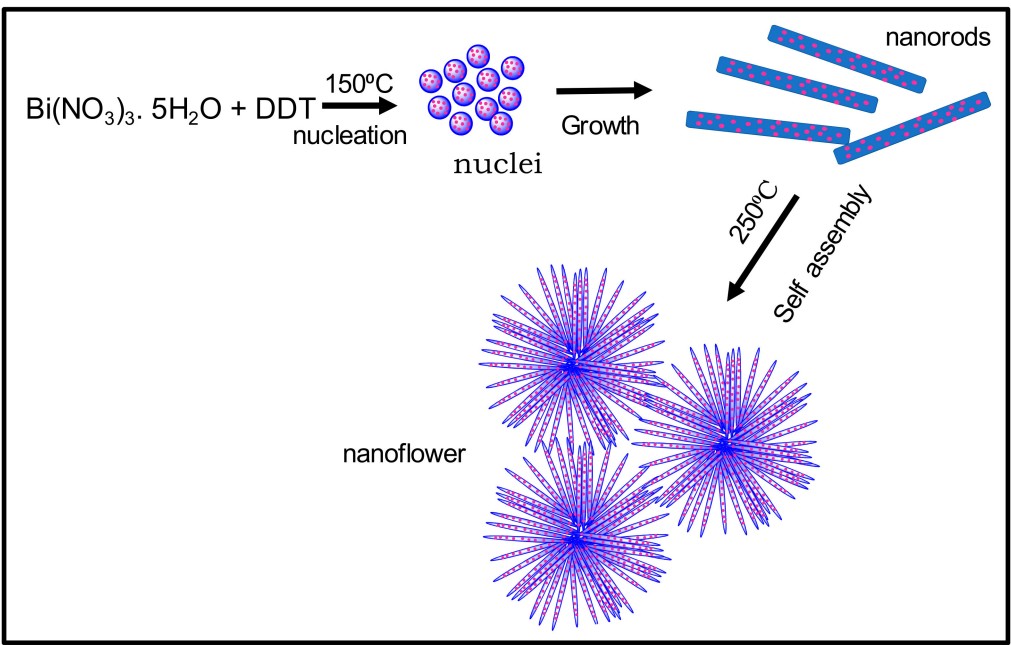

**Figure 1.** Schematic diagram for the synthesis and growth of Bi$_2$S$_3$ nanorods and nanoflowers. DDT-1-docanethiol (DDT).

## 2. Results and Discussion

The colloidal synthesis of the Bi$_2$S$_3$ rod and flower-like nanocrystals was accomplished through the variation in the reaction temperature as shown in Figure 1. The transparent color of DDT darkened upon injection of the Bi$_2$S$_3$ solution and gradually turned black in color.

The morphological changes in the as-prepared materials were analyzed using scanning electron microscopy (SEM), Transmission Electron microscopy (TEM) and High resolution TEM (HRTEM) techniques At 150 °C, the formation of rod-like Bi$_2$S$_3$ nanocrystals with 350–360 nm average length and 65–70 nm average diameter was initiated as revealed by TEM analysis (Figure 2a and Figure S1).

This morphology was preserved until the temperature was increased to 250 °C. Nucleation of AuNPs (average size of 15 nm, Figure S1) on these particles preferentially occur on the surface of these rods, forming BNR-Au (Figure 2b). The loading efficiency of the AuNPs was quantified from the SPR absorption peak of AuNPs before loading and that of the supernatant solution after loading (Figure S2). A loading efficiency of 72% was achieved in the formation of BNR-Au. The lattice fringe (0.55 nm) visible in the HRTEM images (Figure 2c) corresponds to the (001) crystallographic plane of the orthorhombic Bi$_2$S$_3$ structure [29]. When the reaction temperature was increased to 250 °C, the product changed morphology from rod-shaped particles to flower-like particles with sizes between 2 µm and 4 µm (Figure 2d). BNF-Au was obtained by AuNP nucleation as before, effectively decorating the leaf-like surfaces with AuNPs (Figure 2e), which is also seen in EDS elemental mapping (Figure 2f). A loading efficiency of 81% was observed in the BNF-Au. Because separation of charge carriers played a significant role in the photodegradation process, a variation in charge separation was expected in different morphologies due to a variation in the surface-to-volume ratio. Indeed, a significant degradation efficiency in dumb-bell shaped core-shell Au-Bi$_2$S$_3$ particles and Bi$_2$S$_3$ nanoparticles decorated with Au particles has been reported [26]. The enhanced performance by BNF compared to BNR could be due to enhanced charge separation as will be discussed later.

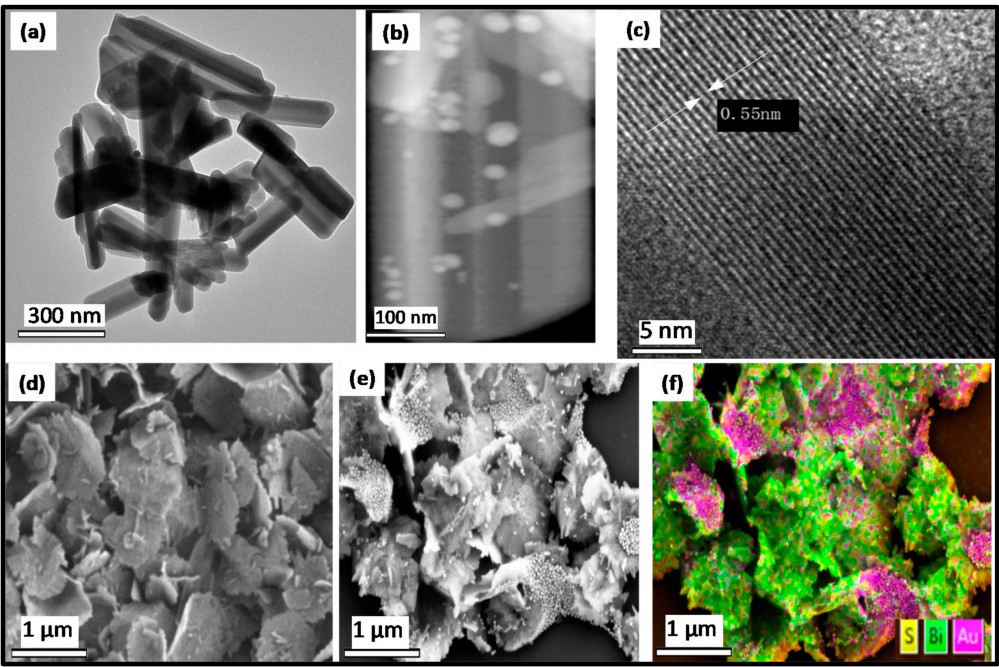

**Figure 2.** (Top) TEM image of rod-shaped $Bi_2S_3$ nanocrystals (BNR) (**a**), bright field TEM for BNR-Au (**b**), HRTEM (**c**) for BNR. (Bottom) SEM images for flower-shaped Bi2S3 nanocrystals (BNF) (**d**), BNF-Au (**e**), elemental mapping for BNF-Au (**f**).

To gain insights into the growth mechanism of the nanocrystals, aliquots from the reactions were drawn at different time intervals during the synthesis at subsequently elevated temperatures. The SEM analysis of the sample drawn after 5 min of bismuth nitrate-DDT solution injection showed nucleation of spherical particles, which quickly morphed into nanorods within 10 min (Figure S3). At 250 °C, the nanorods changed into flower-like nanocrystals. The high-magnification TEM image of single leaf-like flowers showed that the nanorods had assembled into flower-like particles (Figure S3). Thus, the growth mechanism commenced with the nucleation of spherical $Bi_2S_3$ upon injection of the precursors and was followed by the growth and restructuring into nanorods, which then could undergo self-assembly at high temperatures to form flower-like nanocrystals as proposed in Figure 1.

## 2.1. Optical Studies

The absorption spectra of the AuNPs, BNRs, BNFs, and the corresponding gold-decorated analogues (BNR-Au and BNF-Au) are shown in Figure 3a. The optical band gap using Kubelka–Munk transformations plot (Figure S4) for the $Bi_2S_3$ nanostructures was determined to be 1.27 eV. A strong absorption peak is observed at 528 nm and corresponds to the SPR mode for dispersed metallic AuNPs in solution [28]. However, when the AuNPs are nucleated on $Bi_2S_3$ particles, their pronounced optical signature is broadened and is superposed on the absorbance spectrum of the corresponding $Bi_2S_3$ particles. The SPR band broadening is accompanied by a red-shift and is observed around 500–800 nm. The broadening and red shift in the plasmonic peak of AuNPs has been reported and attributed to plasmon–plasmon coupling between adjacent Au nanoparticles, exciton–plasmon coupling interaction, and the higher dielectric constant of the semiconductor $Bi_2S_3$ support as compared to dielectric environment in solution for dispersed AuNPs [30–32].

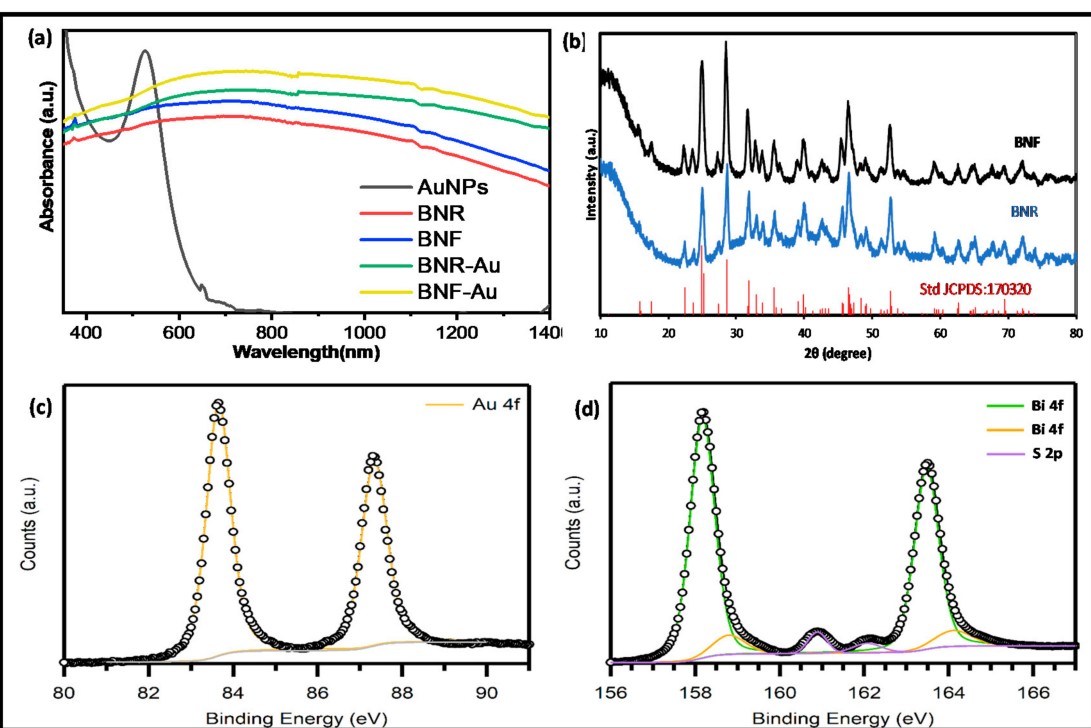

**Figure 3.** UV-vis absorption spectra of the nanocrystals in toluene (**a**), XRD pattern of BNR and BNF (**b**), and the representative high resolution XPS spectra of Au4f, (**c**), S 2p and Bi4f for BNF-Au (**d**).

The crystal structure as well as the lattice planes of the BNRs and BNFs were investigated using powder XRD (Figure 3b). The observed peaks at 2θ = 22.4, 24.9, 28.0, 31.6, 47.5, and 52.4° match the (220), (130), (211), (221), (431), and (351) crystallographic planes of the orthorhombic $Bi_2S_3$ phase (JCPDS:170320) [13]. The well-resolved sharp and narrow diffraction peaks indicate that the nanocrystals exhibit a high degree of crystallinity. The crystallite diameter size was calculated using Debye–Scherrer's (Equation (1))

$$d = \frac{0.89\lambda}{\beta COS\theta} \tag{1}$$

where λ is the wavelength of the X-ray source (λ = 1.5405 Å), β is the full width at half maximum of the diffraction peak, and θ is the angular position.

The sizes were determined by focusing on the (111) plane peaks of the nanoparticle for BNR. The XRD diameter size was determined to be 55 nm and 63 nm BNR and BNF, respectively. The diameter size obtained for BNR was in close agreement with the sizes determined by TEM while significant variation in diameter size was obtained for BNF.

The oxidation states of gold, bismuth, and sulfur in the nanocrystals were identified by high-resolution X-ray photoelectron spectroscopy (Figure 3c,d, using the BNF-Au sample as a representative). The XPS binding energies for the doublet peak of gold are observed at 84.0 eV and 87.1 eV for the $Au4f_{7/2}$ and $Au4f_{5/2}$ peak, respectively, and feature a peak separation of 3.7 eV typical of pure $Au^0$ [33]. The XPS signal deconvolution of Bi 4f revealed the main $Bi\ 4f_{7/2}$ and $Bi\ 4f_{5/2}$ signals with peak separation of 5.3 eV at 158.2 eV and 163.5 eV, respectively, which can be assigned to $Bi^{3+}$ and is characteristic for $Bi_2S_3$ [34].

The EDX spectra (Figure S5) showed an absence of any metallic oxide in the nanostructures. The silicon peak in the spectra arose from a silicon wafer used to prepare the sample. Both experimental and theoretical work have shown that the spin-orbit coupling doublet of $Bi\ 4f_{7/2}$ and $Bi\ 4f_{5/2}$ overlap with that of the S 2p region [35,36] with the binding energy of S 2p laying between the Bi 4f doublets at 160.9 eV and 162.1 eV corresponding to $S\ 2p_{3/2}$ and $S\ 2p_{1/2}$ signals, respectively [32].

### 2.2. Photocatalytic Activity Study

To evaluate the potential photocatalytic application of the prepared nanocomposites for the removal of environmental contaminants, the photodegradation of methyl orange and rhodamine B under simulated solar irradiation was performed. The intensity of the characteristic absorption peak of MO and RhB at 547 nm and 465 nm wavelengths was chosen to monitor the photocatalytic degradation process. Figure 4 shows the evolution of the absorption spectra of the dyes in aqueous solution (initial concentration is $9.27 \times 10^{-5}$ M for MO and $1.72 \times 10^{-5}$ M for RhB, 100 mL) in the presence of 50 mg of the photocatalyst over time. The intensity of dye's absorption peaks drops rapidly with increased irradiation time. The control experiments (i.e., without visible light; Figure S6) showed negligible dye degradation, indicating that the degradation reaction is truly driven by a photocatalytic process.

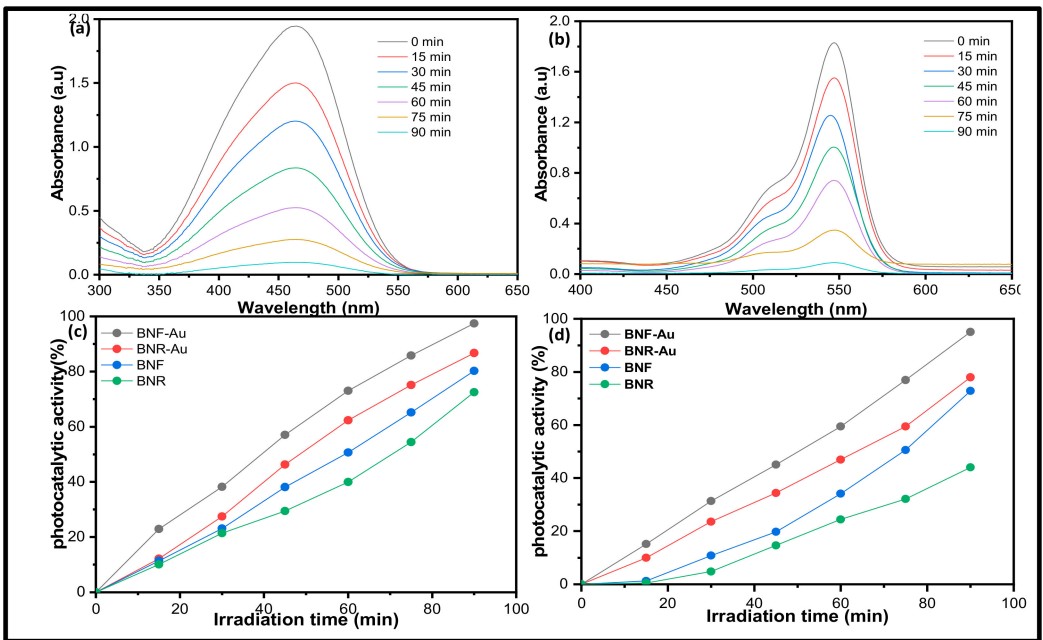

**Figure 4.** Representative absorbance changes of MO (**a**), RhB (**b**) using BNR-Au and the percentages degradation efficiency by the nanocrystals and corresponding gold decorated analogues for MO (**c**) and RhB (**d**).

The photocatalyst degradation efficiency ($\eta$) of each photocatalyst sample was obtained using (Equation (2)) [37]

$$\eta(\%) = \frac{C_0 - C}{C_0} \times 100 \qquad (2)$$

where $C_0$ is the initial concentration of the dyes before light irradiation, and $C$ is the concentration after equal time intervals of light irradiation.

While BNF and BNR exhibit a MO degradation efficiency of 80.2% and 72.5%, the efficiency increased to 97.4% and 86.7% for the heterostructured BNF-Au and BNR-Au (Figure 4c), suggesting enhanced charge carrier injection from the formed heterostructures. Similar observations were also recorded in the degradation of RhB with degradation efficiency for BNF-Au and BNR-Au reaching 95.1% and 77.8% compared to 72.9 and 44.0% for BNF and BNR, respectively (Figure 4d). To confirm the charge separation in the as-prepared nanoparticles, the photoluminescence (PL) was recorded at the excitation of 350 nm. Both BNR and BNF showed a PL peak around 432 nm (Figure S7), which is close to 435 nm obtained by Peng et al. in $Bi_2S_3$ nanospheres [38]. The intensity of the PL peak in BNR is higher than that of BNF, suggesting a better charge separation for the latter. A drastic quenching of the PL signal was observed when the nanocrystals were decorated

with AuNPs (Figure S6, using BNF-Au as a representative), indicating enhanced charge separation through extraction of the electron generated at the conduction band of $Bi_2S_3$ nanoparticles by AuNPs. The pseudo-first order reaction (Equation (3)) [39] was used to obtain the reaction rate constant *k* at time *t*, utilizing the different photocatalysts.

$$In\frac{C_o}{C} = kt \tag{3}$$

The rate constants are found to be 0.018, 0.020, 0.023, and 0.042 $min^{-1}$ for BNR, BNF, BNR-Au, and BNF-Au for the degradation of MO. The corresponding rate constants in the degradation of RhB are 0.006, 0.015, 0.018, and 0.034 $min^{-1}$.

The comparative reaction rate constant for the different nanostructures is shown in Figure 5a. An increase in photocatalytic activity was more than doubled when the nanostructures were decorated with AuNPs compared to BNR or BNF alone.

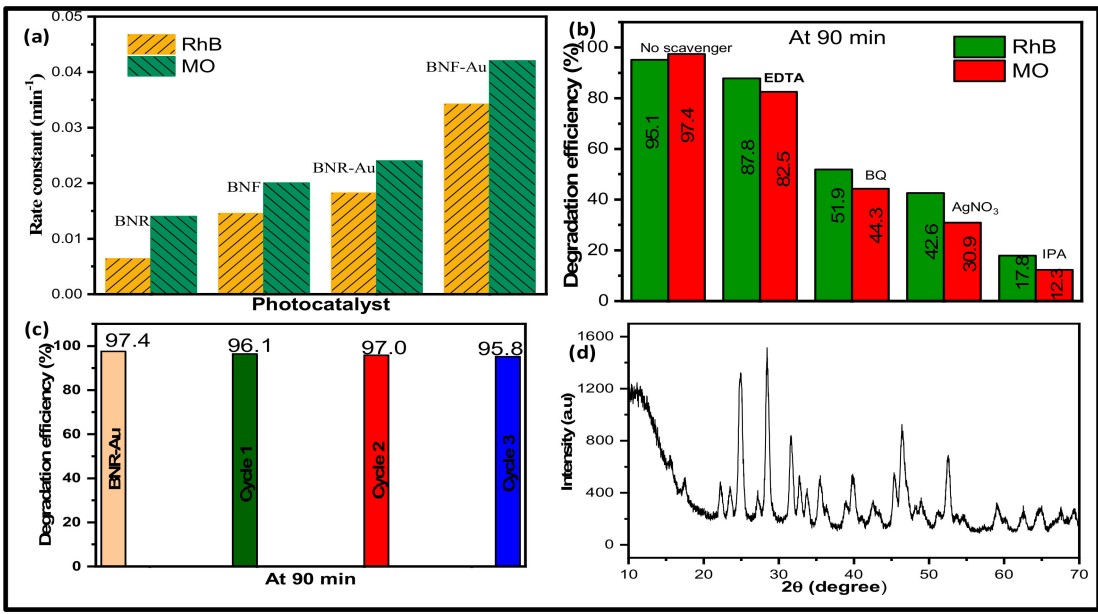

**Figure 5.** (**a**) Comparison of rate constants of the $Bi_2S_3$ nanocrystals and their gold-decorated analogues, (**b**) the effect of different active species-trapping experiments on degradation using BNF-Au as a representative, (**c**) percent degradation efficiency with increasing number of catalytic cycles at irradiation time of 90 min, and (**d**) XRD pattern of BNF after three catalytic cycles.

The role of reactive species responsible for the photodegradation activity was investigated by performing a series of control experiments in the presence of different scavengers such as isopropanol for hydroxyl radicals (OH*), benzoquinone (BQ) for superoxide anion radicals ($O_2^{*-}$), silver nitrate ($AgNO_3$) for electrons ($e^-$), and ethylene diamine tetraacetic acid disodium salt (EDTA-2Na) for holes ($h^+$). A certain amount of these scavengers was introduced into the dye solution prior to the addition of the catalyst and the results were compared to the experiment without scavengers. The final concentration of isopropyl alcohol (IPA), BQ, and EDTA-2Na in the reaction system was 5 mmol $L^{-1}$ each. Utilizing these scavengers, the photodegradation of MO and RhB was suppressed from 97.4% and 95.1% to 30.9% and 42.6% by the addition of $AgNO_3$, 44.3%, and 51.9% by the addition of BQ, respectively, suggesting involvement of electron and superoxide radicals in the degradation process (Figure 5b). A drastic suppression of the photodegradation activity to 12.3% and 17.8% in MO and RhB degradation was observed in the presence of IPA, indicating that hydroxyl radicals dominated the degradation process. From the active species-trapping experiments, it is reasonable to conclude that the hydroxyl radical is the dominant species for the photodegradation with the electron and superoxide performing a significant role. The stability and reusability of the photocatalyst in photocatalytic pro-

cesses is very important to lower the operational cost of their application, thus making photocatalysis with stable materials a captivating approach for wastewater treatment.

To this end, three consecutive cyclic photodegradation experiments were performed using the same recovered catalyst from the preceding tests in fresh dye solutions. A nearly constant photodecomposition rate through three cycles showed that the catalyst could be recycled and was not photo-corroded during the photocatalytic oxidation (Figure 5c). The photostability of the catalysts was further monitored using X-ray diffraction studies during the photocatalytic process. The XRD pattern of the recovered samples revealed the intact phase for the catalyst after three degradation cycles using BNF as a representative sample (Figure 5d). The SEM images after photodegradation experiment for the $Bi_2S_3$ nanostructures (BNF and BNR) alone do not show any obvious change in their morphology, while their Au-BNF and Au-BNR counterparts exhibit shrinkage in the lotus-like flower surfaces (Figure S8). Though the photocatalytic activity improves in our Au-$Bi_2S_3$ heterostructures, the $Bi_2S_3$ alone seems to be more morphologically robust after performing photocatalytic dye degradation tests on them.

### 2.3. Photodegradation Mechanism

Complex optical processes are often involved in nanostructures involving junctions between a semiconductor and a plasmonic nanoparticle. Whether the optical excitation of a plasmon contributes charges for redox chemistry is tedious to disentangle and requires specialized methods such as transient absorption spectroscopy to track charge carrier dynamics across the Schottky barrier at the junction. The channel to drive a chemical reaction through plasmon excitation is dependent on non-radiative decay of the plasmons into hot electrons and holes. Because only plasmons that decay non-radiatively can contribute (radiative decay implies decoupling of the energy from the plasmonic nanoparticle through scattering of a photon), the ratio of scattered and absorbed light in the plasmonic structure could give an upper limit for the efficiency for plasmon-driven processes [40]. On the other hand, metal nanoparticles can serve as traps for charge carriers, which would significantly reduce the photoactivity of a pristine semiconductor nanoparticle. In our system, we employ large $Bi_2S_3$ nanoparticles which can display significant light scattering (Mie scattering), so that the separation of non-radiative and radiative plasmon decay is not trivial. To gain more insights into this and similar material systems, investigation of the total absorption and scattering efficiencies for semiconductor nanoparticles with and without plasmonic nanoparticles could be instructive. Considering the band alignment in $Bi_2S_3$-AuNPs, a possible mechanism for the photocatalytic degradation is shown in Figure 6. The conduction (CB) and valence band (VB) of $Bi_2S_3$ has been reported as −0.1 eV and +1.3 eV vs. NHE, respectively [41], while the Fermi energy level for AuNPs is –0.56 eV vs. NHE [42]. Upon illumination of light, the incident photon is captured by the $Bi_2S_3$, leading to excitation of electrons from the valence band to the conduction band, leaving behind holes in the valence band. Due to the SPR effect, the AuNPs also simultaneously absorb photons from the incident light, which can result in the generation of hot electrons. It was shown that hot electrons can be injected into the conduction band semiconductors, i.e., $Bi_2S_3$ in this case, and contribute to the reduction of $O_2$ to superoxide radicals. The formation of the superoxide anion radical is favorable because the CB potential of $Bi_2S_3$ is more negative as compared to the standard redox potential of $O_2/O_2$*- (−0.046 eV vs. NHE [43], indicating that the transferred electrons in the CB of $Bi_2S_3$ can oxidize $O_2$ into $O_2$*- radicals [44].

Though the holes at the valence band of $Bi_2S_3$ could not possibly produce hydroxyl radicals directly because of the more negative oxidation potential (+1.3 eV vs. NHE) compared to the standard reduction potential of $OH^-/OH$* (1.99 eV vs. NHE [42]), it may oxidize $H_2O$ to produce oxygen ($O_2$) and hydrogen ions ($H^+$), because the VB potential is more positive than the standard oxidation potential of $O_2/H_2O$ (1.23 eV vs. NHE [45]). Furthermore, the CB potential of $Bi_2S_3$ is favorably situated to enable the conversion of $O_2$ into hydrogen peroxide ($H_2O_2$) (+0.69 eV vs. NHE [45]. Thus, the photogenerated

electrons in the CB of $Bi_2S_3$ can undergo multielectron reduction of oxygen and consume the generated $O_2$ and $H^+$ ions species to produce OH* radicals through secondary reactions. These generated hydroxyl and superoxide radicals then decompose the dye molecules. A scheme of the involved energy levels and standard redox potentials is shown in Figure 6 and is consistent with the obtained results from active species-trapping experiments.

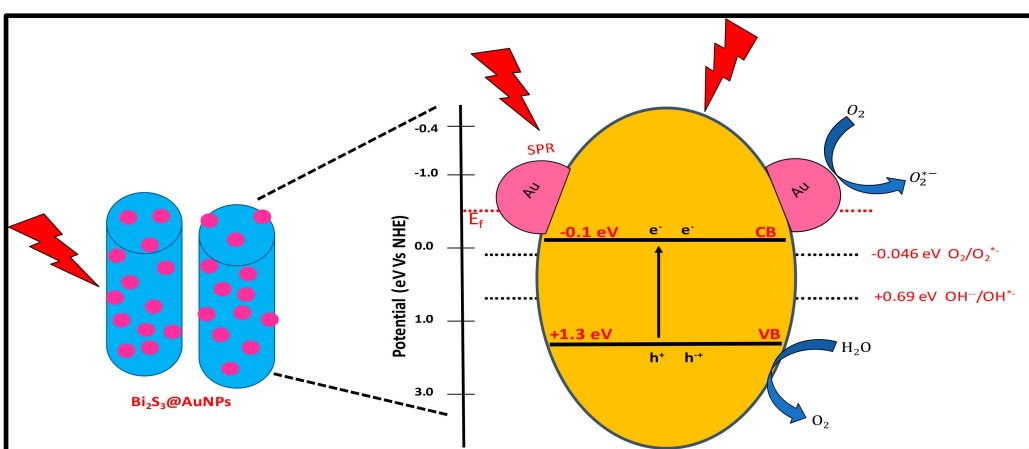

**Figure 6.** Schematic diagram showing possible photocatalytic degradation mechanism for the nanocomposites under visible-light exposure.

## 3. Materials and Methods

Materials: Bismuth nitrate ($Bi(NO_3)_3$ $5H_2O$, Sigma Aldrich, St. Louis, MO, USA, 99.99%), $AgNO_3$ (99.9%, 98%), gold chloride hydrate ($HAuCl_4$. $xH_2O$, 99.99%), dimethy-lammonium bromide (DDAB, 98%) were purchased from Sigma Aldrich, Beijing, China, dodecanethiol (DDT, 98%), dodecylamine (DDA, 98%), octadecene (tech. 90%), isopropyl alcohol (IPA), ethylene diamine tetraacetic acid disodium salt (EDTA-2Na), benzoquinone (BQ) were purchased from Alladdin, Shanghai, China.

### 3.1. Methods

3.1.1. Synthesis of $Bi_2S_3$ Nanorods and Nanoflower Particles

A 30 mL DDT in a two-necked flask was degassed under vacuum in Schlenk line and then gradually heated to 150 °C. $Bi(NO_3)_3$ $5H_2O$ (0.48 g, 1.0 mmol) was dissolved in 5 mL of DDT and quickly injected into the pre-heated solution. The reaction mixture was stirred for 1 h under the flow of argon and subsequently cooled to room temperature. The nanocrystals were collected using isopropanol followed by centrifugation at 8000 rpm for 10 min and subsequently purified by multiple precipitation and re-dispersion with isopropanol and hexane, respectively. The synthesis of $Bi_2S_3$ nanoflowers followed the same procedure except for a change in reaction temperature from 150 °C to 250 °C.

3.1.2. Synthesis of BNR-Au and BNF-Au Particles

A mixture of 50 mg DDAB, 150 mg of DDA, and 30 mg of $HAuCl_4$ was dissolved in 10 mL of toluene and sonicated for 30 min. When the reaction color of the gold precursor solution gradually turned from dark orange to light yellow, the measured absorbance showed the appearance of SPR band of AuNPs. The reaction mixture was added dropwise to the 60 mg dispersion of the as-synthesized nanocrystals in 40 mL toluene solution at room temperature and allowed to stir for 1 h. The heterostructured nanocrystals were precipitated by adding methanol and centrifugation at 8000 rpm for 10 min before final dispersion into toluene. The absorption spectra of the supernatant were recorded and the SPR band of AuNPs in the supernatant and that before loading were used to calculate the percentage amount of AuNPs loaded.

*3.2. Photocatalysis Experiments*

For the photocatalytic characterization, 50 mg of the as-synthesized nanocrystals were dispersed in 100 mL of 20 mg/L aqueous rhodamine B or methyl orange solution and stirred in the dark for 12 h for maximum adsorption of the molecules on the particle's surface. The solution was then transferred to a cubic quartz cell having a cell edge length of approximately 5 cm and a small capped opening at the top. A 300 W Xenon lamp (Newport, Irvine, CA, USA, Model 66902) equipped with an AM1.5 filter placed at a distance of 30 cm was used to irradiate the quartz cell. The light intensity irradiating the cell was measured to be 100 mW/cm$^2$ using a power meter. A separate experiment to confirm various radicals participating in the photodegradation process was performed. In this experiment, 0.5 mg of different sacrificial agents was added with the photocatalyst in the dye solution before illumination. The photodegradation activity was monitored by acquiring the UV-vis spectra of samples taken before, and after, every 30 min irradiation period. A sample volume of 4 mL was drawn every 30 min and centrifuged to remove the catalyst before UV-vis measurement.

*3.3. Instrumentation*

Scanning electron microscopy (SEM) combined with energy dispersive X-ray spectroscopy (EDS) were obtained using a Sigma 500 system (Zeiss and Oxford Instruments). Transmission electron microscopy (TEM) characterization was performed on an America FEI G2 Tecnai instrument operating at 20 kV. The PL measurements were performed on a FL 3-11 steady/transient fluorescence spectrometer at room temperature. Sample was prepared by dissolving 1 mg each of the as-synthesized nanostructures in 10 mL of toluene and sonicated for 5 min. The solution was drop-cast on a silicon wafer or Cu grid for SEM and TEM analysis, respectively. X-ray photoelectron spectroscopy (XPS) was obtained using a Thermo Fisher Scientific K-Alpha X-Ray photoelectron spectrometer with Al K$\alpha$ = 1486.6 eV excitation. The binding energies were corrected by referencing the C1s line to 284.80 eV. Powder X-ray diffraction (XRD) patterns were acquired using a Bruker D8 Advance diffractometer with Cu K$\alpha$ radiation. UV-vis absorption spectra were acquired with the use of a PerkinElmer Lambda 750 spectrophotometer.

**4. Conclusions**

In summary, rod and flower-like shaped orthorhombic Bi$_2$S$_3$ photocatalysts were successfully synthesized and decorated with gold nanoparticles using an efficient colloidal wet chemical synthetic approach. A possible growth mechanism was proposed to explain the formation of the orthorhombic phase nanocrystals. These nanocrystals and the heterostructured derivatives display high photocatalytic activity under solar irradiation for the degradation of organic pollutants. Analysis of the active species-trapping was used to disentangle the reaction mechanism of the dye photodegradation. The particles showed stability and recyclability, rendering them attractive for practical application. Moreover, the synthetic route is a simple and economical suitable for a scale-up process and provides guidance for the design and fabrication of advanced photoactive materials for catalysis and other applications.

**Supplementary Materials:** The following are available online at https://www.mdpi.com/2073-4344/11/3/355/s1, Figure S1: TEM and size distribution for the AuNPs, Figure S2: Absorption spectra of AuNPs before (blue) and the supernatant (super.) solution after forming heterostructured BNF-Au and BNR-Au, Figure S3: SEM images of aliquot of the reaction mixture, Figure S4: The Kubelka–Munk transformations plot of ($\alpha$hv)$^2$ versus photon energy, Figure S5: Representative EDX spectra of BNF and BNF-Au, Figure S6: Absorbance changes of RhB without light after 12 hr. adsorption of the photocatalyst, Figure S7: Photoluminescence spectra of nanostructures recorded at excitation wavelength of 350 nm, Figure S8: SEM images of BNF and BNF-Au before and after degradation experiment.

**Author Contributions:** Conceptualization, original draft preparation, methodology N.N.; validation, formal analysis E.M.A.; supervision, funding acquisition, M.G. All authors have read and agreed to the published version of the manuscript.

**Funding:** The research leading to these results has been funded by the Guangdong Innovative and Entrepreneurial Team Program (no. 2016ZT06C517) and the Australian Government through the Australian Research Council Grant CE170100026. E.M.A. acknowledges funding by the Alexander von Humboldt Foundation through a Feodor Lynen Research Fellowship.

**Data Availability Statement:** Not applicable.

**Conflicts of Interest:** The authors declare no conflict of interest.

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
