# Peer review of "Gold Nanoparticle-Decorated Bi2S3 Nanorods and Nanoflowers for Photocatalytic Wastewater Treatment"

_catalysts, doi:10.3390/catal11030355_

Round 1

Reviewer 1 Report

The authors report on Bi2S3 semiconductor particles functionalized with gold nanoparticles for photocatalytic applications. Two morphologies are distinguished, nanorods and nanoflowers. The latter are described as formed by assembly, which is not fully clarified. The photocatalytic activity was studied by means of organic dyes using solar irradiation. The hybrid samples of Au/Bi2S3 showed enhanced photodegradation capabilities, which was attributed to the release of O2- radicals. The underlying mechanism was attributed to the formation of hot electrons, another not fully proven point. In summary, I believe that this manuscript is an excellent fit for the scope of the journal Catalysts. However, I doubt that the scientific discussion will reach the high standards of the journal. This could certainly be improved in a revision, which I thus encourage. Therefore, I evaluate this submission to require major revisions. In addition, I would like to express that the presentation quality is relatively low and should be improved.

Comments/questions:

  1. The authors should briefly discuss the impact of morphology. In fact, the photodegradation is primarily mediated by the available surface area. As such, one could expect that the morphology of the heterocatalysts does not matter much. At the same time, the band gap should be independent from the morphology of the Bi2S3, right? What is the main hypothesis that motivated to study different morphologies? Please clarify in the manuscript.
  2. The emission spectrum of the solar-like illumination should be reported in the SI.
  3. The experimental procedures applied for loading of Bi2S3 with noble metals is not clear. Please clarify. Maybe it would help to extend Fig. 1 to include the AuNP loading.
  4. The authors might want to discuss an alternative approach to use gold nanoparticles as core and to decorate them with small Bi2S3 nanocrystals to generate core/satellite morphologies or similar colloidal superstructures. Also, it might be beneficial to use preformed AuNPs (possibly of anisotropic shapes such as rods or triangles) because of their high absorbance cross-sections which might increase the formation of hot carriers significantly. I suggest to discuss this alternative approach in brief and to compare the respective benefits and challenges involved.
  5. The authors state that “strong light scattering” as a motivation to use AuNPs. However, small quasi-spherical AuNPs of 5-20 nm in diameter show almost exclusively light absorption with very low scattering contributions. As such, describing them as source of “strong light scattering” is quite misleading. For more details, please refer to basic Mie theory.
  6. The UV/vis spectra are giving a limited insight into the (complex) optical properties of the presented samples. This is at least is part given by the fact that the experiments are performed in classical transmission geometry. As a suggestion, diffuse reflectance spectroscopy, this is a UV/vis spectrometry performed using an integrating sphere, allows to distinguish between the absorbance and scattering contributions (as recently described in DOI: 10.1021/acsami.0c16398). The gold nanoparticles could be expected to strongly contribute to light absorption (non-radiative plasmon decay) and the Bi2S3 might show a significant (Mie) scattering contribution simply by their comparatively large size. I suggest to briefly discuss this option in the manuscript and for future experiments.
  7. The authors state that no significant changes could be observed in the XRD spectra even after several degradation cycles. Could this be caused by XRD mainly probing the bulk of the sample? Changes by degradation would mainly affect the surface. Is XRD sensitive to the surface and if yes, to which extend does it allow to judge material changes by degradation?
  8. The photodegradation mechanism is not completely clear. The degradation of a dye is only an indirect proof. Is there any experimental way to prove the formation of O2- radicals in a direct way?

Author Response

Thank you for reviewing our manuscript. The issues raised and suggestions has been addressed and attached herein.

Reviewer 2 Report

This work presents synergistic harmony of Bi2S3 nanostructures and Au nanoparticles for enhanced photocatalytic activity in the dye degradation. The conclusions drawn by the authors are well supported by the data and fit to the scope of Catalysts. However, there are so many typos and grammatical errors in the entire manuscript, which are disturbing. Before acceptance, they should be corrected. Here are a few examples:

  1. Title: Treatmente to Treatment
  2. Line 87-88: mixed use of american English and British English (color and colour)
  3. Figure 4: the y-axis is not arbitrary unit (a.u.), but just "Absorbance

There are a lot more...

Author Response

Thank you for your expert review. The suggestions and issues raised has been addressed . find attached.

Author Response

Thank you for your expert review. The suggestions and issues raised has been addressed and attached herein

Reviewer 4 Report

The manuscript entitled "Gold Nanoparticle-Decorated Bi2S3 Nanorods and Nanoflowers for Photocatalytic Wastewater Treatment" is very interesting because Bi-based photocatalysts are promising candidates for photocatalytic environmental recovery using visible light. I think that the manuscript deserves to be published. However, I have some questions for the authors:

  1. The introduction should focus on similar work and show the novelty of your research.

https://doi.org/10.1002/anie.201402709

https://doi.org/10.1039/C9TC00759H

https://doi.org/10.1021/acs.langmuir.6b03213

 https://doi.org/10.1002/adfm.201403398

  1. How about oxidative photocorrosion? It is well known that photogenerated charges of both types (holes in the valence band and electrons in the conduction band) contribute to this process. The holes will oxidize S (II) ions to S0, and the O2•- radical anions will oxidize S0 to sulfate anions.
  2. How light intensity was measured?
  3. It is necessary to show the full range of XPS. Why are XPS spectra presented for BNF-Au only. Considering the differences in the photocatalytic activity of BNF-Au and BNR-Au, it would be logical to present the XPS spectra for BNR-Au as well.
  4. L 155-157 It is necessary to include data from control experiments in the graphs. It is also necessary to show how the concentration of dyes changed after 12 hours of dark adsorption. These effects should be distinguished.
  5. L 183-184 It would be a good description of this part of the experiment, in the 3.2 (L 264-…). How much scavengers was added?
  6. L181 Benzoquinone (BQ) also reacts with hydroxyl radical and the use of a sole superoxide radical inhibitor is controversial.
  7. I see an amazing similarity of the BNF SEM images before and after the degradation experiment. Unfortunately, such errors raise doubts about the correctness of the rest of the results. I hope this is a mistake by the authors, it needs to be fixed! I recommend to replace a picture or delete this part! 
  1. Figure 3d shows B 4f, probably meant Bi 4f
  2. I would like to understand what is the reason for the differences in the efficiency of degradation of dyes using BNF and BNR. Nothing is said about this in the discussion.I think the readers will also be interested to understand this.

Author Response

Issues raised has been addressed in the attachment

Round 2

Reviewer 1 Report

The authors have revised the manuscript and responded, albeit very briefly, to the questions raised. Overall, I agree with the amendments and even if, from my point of view, some concerns were dealt with very superficially. This is shown by the example of comments 4 and 6. The answers seem understandable, but I do not see any revision or expansion of the manuscript.  I am still convinced that it would be in the interest of the wider readership to mention and discuss these points in the manuscript. I am convinced that this is also in the interest of the authors. 

Regarding comment 4: I agree. Please be so kind to also discuss this aspect in the manuscript.

Regarding comment 6: The authors correctly acknowledged that complex optical processes may arise from the synergistic interaction of Bi2S3 and Au. Thus, I suggest to discuss this in more detail in the manuscript and to explain options for their analysis. Here, diffuse reflectance spectroscopy, which allows to distinguish between the absorbance and scattering contributions (see DOI: 10.1021/acsami.0c16398) might be an excellent tool. Please discuss the interplay of light absorption (non-radiative plasmon decay) and (Mie) scattering contributions, and how these could be experimentally studied.

Author Response

Thank you for the insightful review. We have addressed the comments and suggestions accordingly.

Reviewer 3 Report

See the attachment. 

Author Response

We appreciate the time taken to review the manuscript and valuable suggestions to make the manuscript better. We have address all comments point by point and attached herein.

Reviewer 4 Report

Comments are in attached file.

Author Response

(The authors gave the same response as above.)

Round 3

Reviewer 3 Report

The manuscript can now be accepted. 

Please correct the following before print- 

L158: The XRD diameter size of was determined to be 55 nm and 1.8 μm for BNR and BNF ...." 

Include instrument detail of photoluminescence (PL) characterization

Author Response

We thank the reviewer for taken out time to review this manuscript. The pointed error has been corrected. Instrument for PL study has been added in L348.

Reviewer 4 Report

The authors took into account all the comments and made changes to the manuscript. I agree with the new version. I believe that the manuscript deserves publication in the journal Catalysts.

Author Response

We thank the reviewer for excellent job in the reviewing of the manuscript. We have checked and corrected all possible typographic errors.